# Three-Dimensional Pore Structure Characterization of Bituminous Coal and Its Relationship with Adsorption Capacity

**DOI:** 10.3390/ma16165564

**Published:** 2023-08-10

**Authors:** Bingyi Jia, Shugang Li, Kui Dong, Haifei Lin, Bin Cheng, Kai Wang

**Affiliations:** 1School of Safety Science and Engineering, Xi’an University of Science and Technology, Xi’an 710000, Chinalhaifei@163.com (H.L.);; 2Xi’an Research Institute of China Coal Technology and Engineering Group Corp, Xi’an 710000, China; 3Department of Geoscience and Engineering, Taiyuan University of Technology, Taiyuan 030000, China; dongkui1011@163.com; 4Chongqing Energy Investment Group Technology Co., Ltd., Chongqing 400000, China

**Keywords:** bituminous, pore structure, three dimensions, roughness, AFM, coal-bed methane (coal gas)

## Abstract

Bituminous coal reservoirs exhibit pronounced heterogeneity, which significantly impedes the production capacity of coalbed methane. Therefore, obtaining a thorough comprehension of the pore characteristics of bituminous coal reservoirs is essential for understanding the dynamic interaction between gas and coal, as well as ensuring the safety and efficiency of coal mine production. In this study, we conducted a comprehensive analysis of the pore structure and surface roughness of six bituminous coal samples (1.19% < *R*_o,max_ < 2.55%) using various atomic force microscopy (AFM) techniques. Firstly, we compared the microscopic morphology obtained through low-pressure nitrogen gas adsorption (LP-N_2_-GA) and AFM. It was observed that LP-N_2_-GA provides a comprehensive depiction of various pore structures, whereas AFM only allows the observation of V-shaped and wedge-shaped pores. Subsequently, the pore structure analysis of the coal samples was performed using Threshold and Chen’s algorithms at ×200 and ×4000 magnifications. Our findings indicate that Chen’s algorithm enables the observation of a greater number of pores compared to the Threshold algorithm. Moreover, the porosity obtained through the 3D algorithm is more accurate and closely aligns with the results from LP-N_2_-GA analysis. Regarding the effect of magnification, it was found that ×4000 magnification yielded a higher number of pores compared to ×200 magnification. The roughness values (*R*_q_ and *R*_a_) obtained at ×200 magnification were 5–14 times greater than those at ×4000 magnification. Interestingly, despite the differences in magnification, the difference in porosity between ×200 and ×4000 was not significant. Furthermore, when comparing the results with the HP-CH_4_-GA experiment, it was observed that an increase in *R*_a_ and *R*_q_ values positively influenced gas adsorption, while an increase in *R*_sk_ and *R*_ku_ values had an unfavorable effect on gas adsorption. This suggests that surface roughness plays a crucial role in gas adsorption behavior. Overall, the findings highlight the significant influence of different methods on the evaluation of pore structure. The 3D algorithm and ×4000 magnification provide a more accurate description of the pore structure. Additionally, the variation in 3D surface roughness was found to be related to coal rank and had a notable effect on gas adsorption.

## 1. Introduction

Coalbed methane (CBM) exploitation is primarily concentrated in anthracite reservoirs, with fewer occurrences in bituminous coal reservoirs [1,2,3]. This discrepancy can be attributed to the fact that bituminous coal is softer, contains more cleats, and exhibits stronger heterogeneity compared to anthracite, thereby impeding CBM production [4,5]. The pores within coal serve as the primary spaces for gas adsorption, constituting approximately 90% of the total gas content within coal pores [6]. Consequently, the desorption and migration behavior of gas in coal are closely linked to the coal pore structure [7,8]. Changes in the characteristics of bituminous coal pores are associated with the second stage of coalification. Therefore, accurately describing the changes in bituminous coal pore characteristics and analyzing the impact of alterations in coal macromolecular structure on these characteristics are essential prerequisites for efficient CBM extraction and safe coal mine production.

Due to the limitations imposed by research conditions and the inherent complexity of coal pores, accurately characterizing coal pore structure has proven to be challenging. However, advancements in technology have facilitated progress in this field. Techniques such as low-pressure gas adsorption (LP-CO_2_/N_2_-GA), He gas adsorption, scanning electron microscopy (SEM), and computed tomography (CT) have emerged as valuable tools for this purpose [9,10,11]. Among these technologies, atomic force microscopy (AFM) has been particularly impactful. It has improved the resolution down to 0.1 nm in scanning electron microscopy, allowing for direct imitation of the surface morphology and nanopores without inflicting damage to the samples [12,13]. Furthermore, AFM provides a three-dimensional (3D) image of the reservoir surface [14,15]. Liu et al. [16] found that LP-N_2_-GA generated a higher percentage of nanopores with a diameter < 4 nm compared to AFM, for both coal and shale samples. However, the results obtained from AFM were deemed more accurate. Combining AFM and SEM observations at the nanoscale, as demonstrated by Li et al. [17], has proven effective in revealing pore structure and mechanical properties in both 2D and 3D dimensions. AFM not only enables the measurement of pore parameters but also provides insights into the surface roughness of coal pores. Since coal pores possess a 3D structure, surface roughness plays a significant role in the adsorption capacity of coal. Nevertheless, there is little research on the relationship between coal surface roughness and coal rank, as well as the correlation between coal surface roughness and coal adsorption capacity.

Despite its utility, AFM technology is not exempt from certain limitations. The accuracy of pore analysis results is directly influenced by the segmentation of AFM images. The “grains” module of the Gwyddion software (Version 2.62) was used to segment particles in AFM images, which include the Edge Detection algorithm, Otsu’s algorithm, Segmentation algorithm, Threshold algorithm, and Watershed algorithm. Notably, the most commonly utilized algorithms, Threshold and Watershed, yield differing results in the context of pore structure analysis. Zhao et al. [15] found the Threshold algorithm to be well-suited for the characterization of pores ranging from 10 to 500 nm, while the Watershed algorithm was better optimized for pores between 1 and 200 nm. Chen et al. [18] found that because the number of AFM scanning data points in an area of any size is 512 × 512, the few data points in the region will affect the accuracy of the software, and they reconstructed the AFM three-dimensional topography to improve the accuracy of pore parameters and surface roughness. Beyond image segmentation, it should be noted that image magnification can also influence the resulting measurements [19]. Nowadays, the impact of magnification on AFM images has not been thoroughly investigated.

In this study, our objective was to gain a comprehensive understanding of the pore structure and heterogeneity of bituminous coal, and elucidate their influence on gas adsorption. To achieve this, we employed low-temperature CO_2_/N_2_ gas adsorption (LT-CO_2_/N_2_-GA) and AFM techniques to evaluate the nanopores present in six coal samples with varying ranks (*R*_o,max_ = 1.19~1.98%). Furthermore, we conducted a detailed analysis of the discrepancies in AFM pore structure calculations resulting from different algorithms and magnification levels (×200 and ×4000). The findings contribute to a better understanding of the complex relationship between coal rank and its physical properties, offering crucial knowledge for the advancement of bituminous coal bed methane exploration and utilization strategies.

## 2. Materials and Algorithms

### 2.1. Experimental Methods

The coal samples were obtained from distinct mining areas, including Liangshuijing Mine in the Shenmu mining area (LSJ), Duerping Mine in the Xishan mining area (DEP), Tunlan Mine in the Xishan mining area (TL), Dafosi Mine in the Huanglong mining area (DFS), Changping Mine in the Jincheng mining area (CP), and Gaohe Mine in the Jincheng mining area (GH). By exploring the geological characteristics of the sampling points and selectively collecting block samples from newly developed coal heading faces. All sample quantities, observations, and descriptions adhered to the respective national standards [20,21]. The proximate analysis and determination of vitrinite reflectance (*R*_o,max_) followed the guidelines stipulated by the respective national standards [22,23]. Low-pressure N_2_ gas adsorption (LP-N_2_-GA) experiments were conducted using an automated gas sorption analyzer (Autosorb iQ-MP, Quantachrome Instruments, Boynton Beach, FL, USA), in strict accordance with the national standards [24,25].

### 2.2. Image Scanning and Preprocessing

#### 2.2.1. Image Scanning

AFM experiments required the argon ion polishing of samples to attain a smooth coal surface. Owing to the brittle nature of the coal sample, it was imperative to secure it in polyester resin. The sample size did not exceed dimensions of 10 mm × 10 mm × 3 mm (see Figure 1). The AFM instrument employed in this experiment was the Bruker Dimension Icon, offering a maximum scanning range of 90 μm × 90 μm × 10 μm and a resolution of 0.15 nm in the lateral direction and 0.04 nm in the vertical direction under the Contact Mode. The 3D AFM used to observe the surface morphology and pore structure is a Dimension Icon AFM (Bruker, Santa Barbara, CA, USA) in the PeakForce QNM (Quantitative Nanoscale Mechanical Mapping) mode. The probe tip had a radius of curvature of 10 nm, and a console rigidity of 5.715 N/m. The images were captured across a magnification range of ×200 to ×4000 nm. The images were acquired over a magnification range from ×200 to ×4000 nm. The experimental procedures are as follows: (1) object sample and select the area that contains the pore structure; (2) the images are first obtained at magnification ×200 (Figure 2a), then at magnification ×4000 (Figure 2b).

#### 2.2.2. Image Denoising

Image denoising aims to minimize errors in atomic force microscopy (AFM) operations. Such errors might include the skew of the sample substrate due to manual handling, the bowl-shaped deformation of the scanning surface prompted by the motion of the AFM probe, and the noise interference occurring during the AFM scanning process. The experimental procedure may introduce abnormal height values in the raw AFM data, and the AFM probe’s scanning movement may cause bowl-shaped deformations in the original data. To correct the aberrant high-value areas and bowl deformations, the Nanoscope Analysis software (Verision 1.40r1) was utilized.

#### 2.2.3. Image Segmentation

Image segmentation determines a reasonable segmentation threshold to obtain the pore function, so as to filter out the pores in the image. It is the key to pore structure analysis, and its segmentation results directly affect accuracy.
(1)$$g(x,y)=\{\begin{array}{cc} 1 & f(x,y)>T\\ 0 & f(x,y)\leq T \end{array}$$
where *T* is the threshold, *f* (*x*, *y*) is the original image, and *g* (*x, y*) is the generated binary image.

(1)Threshold algorithm

Commonly used 2D segmentation algorithms mainly include the Huang, MaxEntropy, Otsu, and Yen algorithms [26,27,28,29]. The comparison of different algorithms shows that the Yen algorithm is the most accurate [19]:(2)$$TC(T*)=\underset{T\in G}{\max}TC(T)$$
where *TC*(*T*) is the total amount of correlation contributed by the pores and coal matrix and is defined as follows:(3)$$TC(T)=2\ln(\sum_{i=0}^{T}p_{i}\times \sum_{i=T+1}^{m-1}p_{i})-\ln(\sum_{i=0}^{T}p_{i}^{2}\times \sum_{i=T+1}^{m-1}p_{i}^{2})$$
(4)$$p_{i}=\frac{f(i)}{M\times N}$$
where *M* × *N* pixels is the size of *f* (*x*, *y*) images that are represented by m gray levels. Let G ∈ {0, 1, …, (*m* − l)} denote the set of gray levels and *f*(*i*), *i* ∈ G be the number of gray-level frequencies of the image *f* (*x*, *y*).

(2)Chen’s algorithm

As reported by Chen et al. [18], according to the characteristics of the 3D image, when projecting the 3D sample surface in the xOy plane coordinate system, the xOy plane is used to cut the sample surface from bottom to top, the pore volume is the volume enclosed by the xOy plane, and the sample surface below the plane, which is defined by
(5)$$V(h)=\{\begin{array}{cc} \iint_{D}h-g(x,y)d\sigma & D\in \left\{(X,Y)|g(x,y)<h,x<\frac{a}{2},y<\frac{b}{2}\right\}h\leq T\\ V(T)+A(h-T) & h>T \end{array}$$
where *V* is the pore volume, m^3^; *A* is the projected area, m^2^; *h* is height, m; a and b are the projected width and length, respectively, m.

### 2.3. 3D Roughness

Two-dimensional (2D) roughness metrics cannot comprehensively and accurately portray the morphology of coal surfaces, prompting researchers to shift from traditional 2D roughness analysis to a more exhaustive assessment using three-dimensional (3D) surface roughness parameters [30,31]. The 3D roughness parameter significantly deviates from the conventional 2D roughness single curve analysis, offering a more accurate reflection of the overall surface topography of the coal. Consequently, AFM’s 3D roughness analysis can precisely quantify the microstructure characteristics of the surface topography across different grades of metamorphic coal.

AFM images were imported into Nanoscope Analysis to calculate the average roughness (*R_a_*), the root mean square roughness (*R_q_*), kurtosis (*R_ku_*), and skewness (*R_sk_*).

*R_a_* represents the average distance of the surface deviation from the datum; the calculation formula for *R*_a_ is as follows:(6)$$R_{a}=\frac{1}{MN}\sum_{i=1}^{N}\sum_{j=1}^{M}\left|Z(X_{i},Y_{j}\right)|$$
where *M, N* are the number of data points on the *X*, *Y* (dimensionless), and *Z* (nm) is the height of each data point.

*R_q_* represents the root mean square of the surface deviation from the datum; the calculation formula for *R_q_* is as follows:(7)Rq=1MN∑i=1N∑j=1MZ^2(Xi,Yj)

*R_sk_* represents the degree of asymmetry of the surface height distribution; the calculation formula for *R_sk_* is as follows [32]:(8)$$R_{sk}=\frac{1}{R_{q}^{3}}\frac{1}{N}\sum_{j=1}^{N}Z_{j}^{3}$$

A positive *R_sk_* value indicates that the distribution is to the right and that there are more areas where the sample surface height is lower than the average; a negative value indicates that the distribution is to the left and that there are more areas where the sample surface height is higher than the average (Figure 3a).

*R_ku_* represents the probability that the surface height value is concentrated on the average height value; the calculation formula for *R_ku_* is as follows [33]:(9)$$R_{ku}=\frac{1}{Rq^{4}}\frac{1}{N}\sum_{j=1}^{N}Z_{j}$$

A positive *R_ku_* value means that the waveform reaches its peak, in which the sample surface height is concentrated at the average value; a negative value means that the waveform is flat, so the surface height of the sample is distributed (Figure 3b).

### 2.4. Theoretical Models

#### 2.4.1. Porosity

Porosity is the ratio of the pore volume of coal to the total volume. The total volume of coal is the sum of the skeleton volume and the pore volume. The calculation formula for porosity (%) is as follows:(10)$$\phi =\frac{V_{P}}{V_{P}+V_{M}}$$
(11)$$V_{M}=S_{R}\cdot \sum_{j=1}^{N}\left(Z_{j}-Z_{\min}\right)$$
where *Φ* is the porosity (%), *V_P_* is the pore volume (nm^3^) measured by the Watershed algorithm in Gwyddion, *V_M_* is the coal skeleton volume (nm^3^), *S_R_* is the real area represented by a single pixel (nm^2^), *N* is the number of pixels in the image, and *Z*_min_ is the minimum height of all data points on the image (nm).

#### 2.4.2. Adsorption Experiments

The isothermal sorption curves of CH_4_ on different coal samples were fitted by Langmuir, Freundlich, and Spis isotherm models [34].

(1)The Langmuir adsorption model is as follows:

(12)$$q=\frac{q_{m}bP}{1+bP}$$
where *q* is the adsorption amount at pressure *P* and temperature *T*; *q_m_* is the maximum adsorption ability; and *b* is the Langmuir constant.

(2)The Freundlich adsorption model is as follows:

(13)$$V=K_{b}P^{m}$$
where *K*_b_ and m are fitting constants, which are related to the size of the adsorbing space and the temperature, respectively.

(3)The Sips adsorption model is as follows:

(14)$$q=\frac{q_{m}{\left(K_{s}P\right)}^{1/n}}{1+{\left(K_{s}P\right)}^{1/n}}$$
where *K_s_* indicates the sips isotherm constant.

The adsorption effective capacity was defined using the following equation [35]:(15)$$q_{e}=\frac{(C_{i}-C_{e})V}{M}$$
where *C_i_* and *C_e_* are the initial and final equilibrium concentrations.

## 3. Results

Table 1 presents the results of the proximate analysis and vitrinite reflectance for the coal samples. The samples include two high-volatile bituminous coals (LSJ, DFS), two medium-volatile bituminous coals (TL, GH), and two low-volatile bituminous coals (DEP, CP).

### 3.1. Characterization of Pore Structure by the LT-CO_2_-GA Experiment

According to the method proposed by Hudot [36], the pores of coal are divided into micropores (<2 nm), mesopores (2–50 nm), and macropores (>50 nm). Table 2 presents the micropore structure parameters derived from the NLDFT model, including specific surface area, pore volume, and minimum pore size within the range of 0.012–0.026 cm^3^/g, 32.027–69.924 m^2^/g, and 0.6–0.9 nm, respectively. Notably, it is evident that low-volatile bituminous (LVB) coal exhibits the highest pore volume (PV) and specific surface area (SSA), followed by medium-volatile bituminous (MVB) and high-volatile bituminous (HVB) coal. Furthermore, *V* and SSA demonstrate a polynomial relationship with increasing coal rank.

Figure 4 illustrates the isothermal adsorption curves of CO_2_ for all the samples. At the same pressure, the adsorption isotherm is relationship with the pore structure of coal. As the degree of metamorphism increases, the overall CO_2_ adsorption exhibits an initial decrease followed by an increase. The coal sample LSJ, characterized by a moderate degree of metamorphism, exhibits the lowest adsorption capacity, while DEP and CP coal samples demonstrate higher adsorption capacities.

### 3.2. Characterization of Pore Structure by the LT-N_2_-GA Experiment

The pore structure parameters obtained from the LT-N_2_-GA experiment are presented in Table 3. The SSA and V of the mesopores were determined using the BJH model. The mesopore SSA for all the samples fell within the range of 0.494–1.056 m^2^/g, with the CP coal sample exhibiting approximately twice the mesopore SSA compared to LSJ. The pore volume of BJH mesopores ranged from 2.005 × 10^−3^ to 4.231 × 10^−3^ cm^3^/g.

The coal sample adsorption isotherms are shown in Figure 5. IUPAC classification distinguishes five hysteresis loops, namely, Type H1, H2, H3, H4, and H5. The pore shapes include slit shape, ink-bottle shape, cylinder shape, etc. The adsorption isotherms of the coal samples are shown in Figure 5 [37], which are classified as Type H2 and Type H3.

Type H2: The adsorption curves of DFS, LSJ, and GH exhibit wide-ranging adsorption loops that closely resemble Type H2 behavior. At relative pressures (P/P0) below 0.8, the adsorption and desorption curves coincide closely. However, as the pressure surpasses 0.44, distinct adsorption loops become apparent. At a relative pressure of 0.83, the adsorption capacity experiences a sudden increase, resulting in a steep curve shape with a pronounced concave appearance. When the relative pressure is close to 1, the sample is close to adsorption saturation. Then, as the relative pressure decreases, the desorption of N_2_ commences. Notably, when the relative pressure exceeds 0.75, the desorption capacity exhibits a steady decline. Within the relative pressure range of 0.75–0.45, the desorption curve displays a distinct inflection point characterized by a rapid decrease in desorption quantity, followed by a stabilization phase. The predominant pore types observed in the samples are primarily open-necked holes or ink-bottle-shaped cavities.

Type H3: The adsorption curves of TL, DEP, and CP exhibit a narrow range of adsorption loops, resembling Type H3 behavior. At relative pressures (P/P_0_) below 0.8, the isothermal adsorption curves exhibit a gradual rise, characterized by an upper convex shape, signifying the transition from monolayer to multilayer adsorption. Subsequently, when the relative pressure surpasses 0.8, there is a rapid increase in gas adsorption capacity due to capillary condensation. In contrast, the desorption curves lack prominent inflection points but exhibit a steep decline in desorption quantity at higher relative pressures (above 0.85), forming a distinct concave shape. Following this, the amount of desorption decreases gradually in an approximately linear fashion. The predominant pore type observed in these samples consists of open slot-shaped pores, composed of non-rigid aggregates of plate-like particles.

### 3.3. Characterization of Pore Structure by AFM

Here we only list ×4000 nm image surface of samples for exhibition (Figure 6). In highly volatile bituminous coal (LSJ, DFS), linear parallel cracks develop, and their extension distance is relatively long. The surface is a banded structure with different widths. In middle-rank bituminous coal, GH and TL, the width of microcracks on its surface increases, and micro-cracks coexist with pores. In anthracite coal, DEP and CP, the surfaces present a fibrous structure, the structure tends to be compact, and the morphology becomes flat.

Cracks with a width of 500 nm are called micro-cracks [38]. A micro-crack is formed by the internal tension caused by physical and chemical changes in coal structure under the influence of temperature and pressure in the process of coalification [39]. Some particles are adsorbed in the cracks; the primary cracks give rise to secondary cracks. In the process of coalification, the side chains and functional groups of organic molecules in coal constantly break and fall off with the increase in temperature and pressure, resulting in a large number of volatile products. But the evolution of coalification does not follow a straight line. When *R*_o,max_ is about 1.3%, the second jump occurs, and a large amount of methane escapes from coal. The high fluid pressure caused by a large amount of fluid generation that is discharged over time is the main internal cause of crack generation. When *R*_o,max_ < 1.3%, the crack in coal increases with the increase in gas content. When *R*_o,max_ = 1.3%, the gas generation reaches its maximum, and the crack density in coal reaches its maximum at this stage (DFS and LSJ have obvious cracks). When *R*_o,max_ > 1.3%, the internal tension caused by devolatilizing decreases with the decrease in the amount of merperance, and the newly generated cracks are less. Meanwhile, under the action of increasing in situ stress, the existing cracks gradually close and disappear (the cracks of GH, TL, DEP, and CP become less and less). Therefore, when *R*_o,max_ > 1.3%, the density of cracks in coal decreases with the increase in metamorphism. Therefore, with the increase in metamorphism degree in bituminous coal, cracks in the coal gradually decrease. The crack is the foundation of fracturing, so the development of cracks is conducive to the generation of fractures after fracturing.

## 4. Discussion

### 4.1. Comparing Different Algorithms and Magnification Results of AFM

Table 4 and Table 5 present the statistical analysis of pore structure using different algorithms. It is important to note that the choice of algorithm and magnification can have a notable impact on coal pore volume and surface roughness. The Threshold and Chen’s algorithms exhibit significant influence on the results, while variations in magnification (×200 and ×4000) also yield different outcomes.

At an image magnification of ×200, the Threshold algorithm yields a range of pore numbers between 127 and 264, while Chen’s algorithm results in a range of pore numbers spanning from 154 to 288. On the other hand, at an image magnification of ×4000, the Threshold algorithm produces a pore number range of 494 to 1056, whereas Chen’s algorithm generates a pore number range of 563 to 1220. Whether the analysis is based on the Threshold algorithm or Chen’s algorithm, the smallest pore sizes characterized at ×4000 magnification are smaller than those observed at ×200 magnification. For instance, using the Threshold algorithm, the smallest pore sizes for DFS are 0.18 nm and 0.35 nm at ×4000 and ×200 magnifications, respectively. Similarly, with Chen’s algorithm, the smallest pore sizes for DFS are 0.12 nm and 0.28 nm at ×4000 and ×200 magnifications, respectively. However, there is not much variation observed in the maximum pore sizes characterized between different magnifications. For example, with the Threshold algorithm, the maximum pore sizes characterized are 220.4 nm and 219.2 nm at ×4000 and ×200 magnifications, respectively. Similarly, using Chen’s algorithm, the maximum pore sizes are 234.5 nm and 235.3 nm at ×4000 and ×200 magnifications, respectively. The porosity obtained by calculations based on different algorithms is shown in Figure 7.

The discrepancy between the results obtained from the Threshold and Chen’s algorithms can be attributed primarily to variations in pore shape. The presence of numerous ink-bottle-shaped pores, characterized by a small orifice and a larger throat, leads to smaller pore size measurements with the Threshold algorithm compared to Chen’s algorithm. Regarding the disparities between ×200 and ×4000 images, they arise primarily from the inherent heterogeneity of coal. The broader scope of the AFM image captures a more pronounced manifestation of this heterogeneity. It is important to note that the absolute deviation between ×200 and ×4000 images is irregular, indicating that the strength of heterogeneity characteristics is contingent on the specific region selected within the AFM image.

Figure 7 presents the porosity values obtained using different methods. The porosity distribution obtained from the Threshold algorithm ranges from 7.2% to 19.1%, while Chen’s algorithm yields a porosity range of 7.4% to 21.1%. Conversely, the porosity values determined through LT-N_2_-GA measurements fall within the range of 8.7% to 21.5%. Notably, the results obtained from Chen’s algorithm exhibit a closer resemblance to the porosity values derived from LT-N_2_-GA measurements. Moreover, the difference in porosity between ×200 and ×4000 nm magnifications is similar, suggesting that the effect of magnification on porosity calculations can be considered negligible.

Table 6 provides the values of surface roughness for different magnifications. At ×200 magnification, the *R*_sk_ values range from −0.73 to 0.19, the *R*_ku_ values range from 4.35 to 13.1, the *R*_q_ values range from 52.2 to 132, and the *R*_a_ values range from 30.7 to 107. On the other hand, at ×4000 magnification, the *R*_sk_ values range from −0.31 to 0.19, the *R*_ku_ values range from 2.74 to 5.51, the *R*_q_ values range from 3.85 to 21.1, and the *R*_a_ values range from 2.60 to 16.23. Notably, the *R*_q_ and *R*_a_ values at ×200 magnification are 5–14 times higher than those at ×4000 magnification, with the largest difference observed in the LSJ sample. This can be attributed to the increased variation in height from the mean line as magnification increases, indicating a rougher outer surface. The differences in *R*_sk_ and *R*_ku_ values between ×200 and ×4000 magnifications are relatively small. However, the *R*_sk_ and *R*_ku_ values at ×4000 magnification are generally higher, suggesting a greater prominence of surface peaks at lower magnifications.

### 4.2. Pore Structure Evolution on the Second Coalification Jump

In the LT-N_2_-GA images, high-volatile bituminous coals and middle-volatile bituminous coals exhibit a higher presence of cylindrical pores, while low-volatile bituminous coal displays a significant proportion of wedge-shaped pores. On the other hand, the AFM images reveal distinct characteristics for each coal type. In high-volatile bituminous coal, linear parallel cracks are prevalent, extending over considerable distances, and the surface exhibits a banded structure with varying widths. Middle-volatile bituminous coal exhibits an increase in the width of microcracks on the surface, along with the coexistence of microcracks and pores. Low-volatile bituminous coal displays a fibrous surface structure that tends to be compact and flattened in morphology. By combining the LT-N_2_-GA and AFM images, it becomes apparent that the pore structure of bituminous coal gradually narrows as the degree of metamorphism increases. Furthermore, larger cracks diminish while the presence of micro-cracks becomes more pronounced.

The pore volume, specific surface area (SSA), pore number, pore size, and porosity can be effectively determined through LT-CO_2_-GA, LT-N_2_-GA, and AFM experiments. These pore characteristics exhibit a close relationship with coal ranks. The SSA, PV, and pore number demonstrate a rapid decline within the *R*_o,max_ = 1.19%~1.29%, followed by a gradual increase within the *R*_o,max_ = 1.37%~1.97%. However, the observed values for pore size and porosity differ slightly. The maximum pore size exhibits irregular changes, while the minimum pore size increases between *R*_o,max_ = 1.19%~1.37% before gradually decreasing. Conversely, the variation in porosity follows an opposite trend, wherein it rapidly decreases within the *R*_o,max_ = 1.19%~1.37% and then increases.

The observed phenomenon can be attributed to the changes in the macromolecular structure during the second coalification jump. Low metamorphic coal exhibits an irregular molecular structure, with long side chains and numerous functional groups, leading to the formation of a relatively loose spatial structure with significant micropores SSA and PV. As coal rank increases, the presence of oxygen-containing functional groups and alkyl side chains decreases, while aromatic nuclei increase. This results in a compression of the coal structure, leading to the lowest values of SSA and PV at this stage (*R*_o,max_ = 1.19%~1.29%). With further coalification, a substantial number of aromatic layers are formed, enhancing the ordering of macromolecules and causing the aromatics to be arranged more closely. This arrangement leads to the formation of new cracks, resulting in an increase in micropores SSA and PV, and a decrease in pore size (*R*_o,max_ = 1.37%~1.97%). It is important to note that the changes in porosity and micropore characteristics are not identical.

This discrepancy can be attributed to the stage between *R*_o,max_ = 1.29%~1.6%, where the length of aliphatic chains decreases, leading to an increase in the aromatic system. However, the dehydrogenation of aromatic groups prevents the formation of a well-defined parallel structure in the enlarged aromatic system. As a result, the coal skeleton volume increases, causing a lag in porosity changes compared to micropore changes. Nonetheless, as the aromatic structure gradually arranges itself in a regular manner, the spacing between layers decreases, resulting in increased SSA, PV, and porosity.

### 4.3. Surface Roughness Evolution on the Second Coalification Jump

Surface roughness changed with the degree of coalification, relationships between surface roughness (*R*_a_, *R*_q_, *R*_sk_, and *R*_ku_) and thermal maturity (*R*_o,max_) of naturally matured coals are shown in Figure 8.

As illustrated in Figure 8a,b, the evolution trends of *R*_a_ and *R*_q_ are clearly similar. As *R*_o,max_ = 1.19–1.39%, *R*_a_ and *R*_q_ decrease with increasing *R*_o,max_, with minimum values at *R*_o, max_ = 1.39%; as *R*_o,max_ = 1.39–1.80%, *R*_a_ and *R*_q_ increase with increasing *R*_o,max_ slowly, and more rapidly when *R*_o,max_ = 1.8–2.5%. As illustrated in Figure 8c,d, no significant correlation was observed between *R*_sk_ or *R*_ku_ and *R*_o,max_, suggesting that *R*_o,max_ does not exert control over either *R*_sk_ or *R*_ku_. It is noteworthy that for GH and TL, the major surface heights are below the average (*R*_sk_ < 0), while for other samples, more surface heights are above the average (*R*_sk_ > 0). Additionally, all samples demonstrated *R*_ku_ > 0, indicating a concentration of all sample surface heights around the average value.

The surface roughness of coal can be influenced by both its composition and nanopore development. In coals with relatively low thermal maturity (*R*_o,max_ < 1.3%), the surface roughness is primarily controlled by micro-composition and mineral composition. However, in coals with higher maturity (*R*_o,max_ > 1.3%), nanopore development plays a more dominant role [40]. Thus, the observed variation in surface roughness of bituminous coal with increasing coal rank aligns with the trend observed in nanopore development, wherein it initially decreases and then increases.

### 4.4. Effect of the 3D Pore Structure on CH_4_ Adsorption Capacity

To investigate the relationship between the 3D pore structure and the adsorption capacity of CH4, HP-CH4-GA was carried out on different samples. The Langmuir, Freundlich, and Sips adsorption models were employed to simulate the adsorption isotherms, as illustrated in Figure 9. The isotherm model constants and nonlinear regression parameters are summarized in Table 7. The adsorption isotherm data were analyzed using three different models: Langmuir, Freundlich, and Sips. Among these models, the Sips model demonstrated a strong fit to the experimental data, exhibiting a higher correlation coefficient (*R*_2_ = 0.9958). According to Milan et al. [35], the Sips model is well-suited for capturing the adsorption site interactions that occur on the heterogeneous surface of the adsorbent. In contrast, the Langmuir model assumes monolayer adsorption on a homogeneous surface. Given these considerations, the Sips model is considered more appropriate for accurately characterizing the isothermal adsorption behavior of bituminous coal. The *Q*max of CH_4_ was 5.30~25.12 cm^3^/g according to the Sips adsorption model.

The adsorption effective capacities of the samples are presented in Figure 10. The results indicate that the adsorption capacity of bituminous coal exhibits a trend of initially decreasing and then increasing with increasing coal rank, and it also increases with higher pressure. The order of adsorption capacity from highest to lowest is CP > DEP > TL > GH > DFS > LSJ. The observed ordering of adsorption capacities aligns with the ordering of the *Q*max.

Previous studies commonly suggest that the gas adsorption capacity of coal is influenced by micro- and mesopores [41,42,43]. However, the analysis of pore structure in this study reveals that there is no evident linear relationship between porosity, micropore volume, or specific surface area and coal reservoir adsorption capacity, particularly for lignite and bituminous coal [44,45]. Notably, in this study, the micropore volume and specific surface area of DFS (43.354, 0.019) and TL (48.049, 0.019) are comparable. However, the *Q*_max_ and effective adsorption capacity of TL (9.98, 91%) are higher than those of DFS (17.95, 84%). Similarly, although the micropore volume and specific surface area of GH (43.354, 0.019) are smaller than those of TL (48.049, 0.019), the *Q*_max_ and effective adsorption capacity are comparable. These findings suggest that additional factors beyond micropore volume and specific surface area contribute to the adsorption capacity of coal reservoirs. Functional groups and surface roughness are recognized as factors that can influence adsorption capacity. While there have been numerous studies investigating the impact of functional groups on adsorption, the relationship between surface roughness and coal adsorption capacity has received relatively less attention.

Generally, it has been observed that samples exhibiting a lower surface roughness tend to have a smaller specific surface area and gas adsorption capacity, whereas samples with a higher surface roughness exhibit a larger specific surface area. This increased surface roughness provides more space for gas adsorption [46,47]. The relationship between 3D surface roughness and *Q*_max_ for all samples is presented in Figure 11. The *R*_sk_ and *R*_ku_ values of the samples exhibit a positive correlation with *Q*_max_, as they represent the degree of fluctuation in the sample’s storage space, reflecting the combined surface area and volume characteristics. According to the definitions of *R*_sk_ and *R*_ku_, a greater fluctuation in coal results in a higher gas storage capacity. Therefore, *R*_sk_ and *R*_ku_ can be used as indicators of gas adsorption capacity. Similarly, the values of *R*_a_ and *R*_q_ also show a positive correlation with *Q*_max_. Smaller values of *R*_a_ and *R*_q_ indicate a smoother coal surface, which leads to reduced friction between the gas and coal surfaces. This reduction in friction weakens the intermolecular forces between the gas and coal, resulting in stronger gas adsorption capacity. These findings highlight the significant influence of *R*_a_, *R*_q_, *R*_sk_, and *R*_ku_ values on the analysis of gas adsorption volume.

## 5. Conclusions

Upon conducting a comparative analysis of AFM images across various algorithms and scales, we posit that the pore calculation results derived from the 3D algorithm at ×4000 magnification are more accurate than those obtained through other algorithms. These results exhibit greater resemblance to the LP-CO_2_/N_2_-GA findings. Chen’s algorithm discerned a larger number of pores than the Threshold algorithm. For example, in the case of DFS, the numbers were 1220 (×4000) versus 1056 (×4000). Furthermore, Chen’s algorithm uncovered more micropores. The porosity determined by the 3D algorithm outperformed that of the Threshold algorithm and was closer to the LP-N_2_-GA results. When observed at a magnification of ×4000, more pores were identified than at ×200 (DFS: 1056 vs. 264 using the Threshold algorithm). However, the porosities observed at magnifications of ×200 and ×4000 nm were similar, rendering the effect of magnification inconsequential.AFM, employing Chen’s algorithm and a magnification of ×4000, can accurately analyze the 3D pore structure of bituminous coal. Based on this, the range of pore quantity in bituminous coal is found to be 563–1220, with the maximum value of CP and the minimum of DF. The range of the maximum pore size is 234.5–234.5 nm, while the range of the minimum pore size is 0.12–0.15 nm. These values show minimal variation with respect to coal rank. The variation range of porosity is 7.4% to 21.1%, with GH having the minimum value; *R*_sk_ ranges from −0.31 to 0.19, and *R*_ku_ ranges from 2.74 to 5.51, with weak regularity in their variations. The range of *R*_q_ is 3.85–3.85, and Ra ranges from 2.60 to 17.8, with LSJ having the minimum value and CP having the maximum value. Among the different adsorption models, the Sips model exhibits the best fitting performance. The *Q*_max_ values for CH_4_ adsorption range from 5.30 to 25.12 cm^3^/g. The ordering of adsorption capacity from highest to lowest is CP > DEP > TL > GH > DFS > LSJ, which aligns with the observed ordering of *Q*_max_.The second coalification transition exerts a significant impact on the coal’s pore structure. Over time, the structure evolves from linear, parallel cracks and cylindrical pores to microcracks and wedge-shaped pores. Simultaneously, coal’s pore volume and surface roughness initially decline before escalating, correlating with the coal rank. *R*_a_ and *R*_q_ decrease linearly with the increase; the *R*_ku_ value increases, and the *R*_sk_ value is greater than 0 in the early stage and gradually becomes less than 0. This variation is predominantly attributed to the transformation of the coal’s macromolecular structure.Surface roughness significantly impacts the gas adsorption capacity of samples. A more pronounced fluctuation in coal structure corresponds to a higher gas storage capacity. As a result, *R*_sk_ and *R*_ku_ serve as reliable indicators of gas adsorption potential. Furthermore, smaller *R*_a_ and *R*_q_ values, indicative of a smoother coal surface, result in diminished friction between the gas and coal surface, thereby enhancing gas adsorption.

## Figures and Tables

**Figure 1 materials-16-05564-f001:**
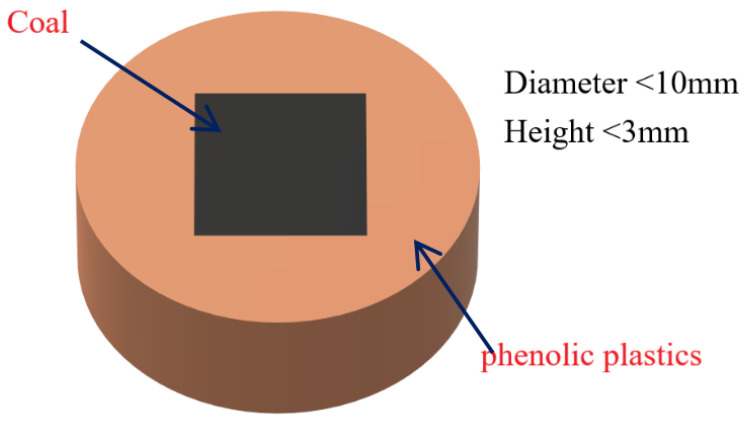
Coal samples.

**Figure 2 materials-16-05564-f002:**
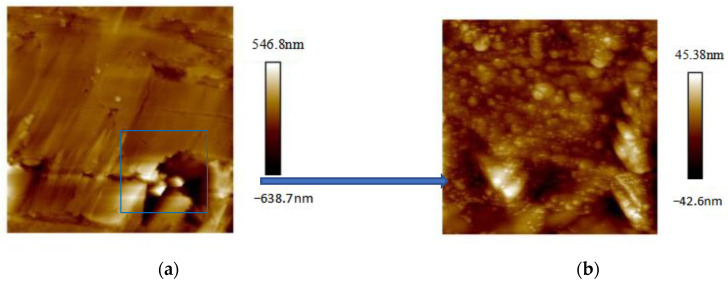
The experimental procedures of the sample: (**a**) image at magnification ×200; (**b**) image at magnification ×4000.

**Figure 3 materials-16-05564-f003:**
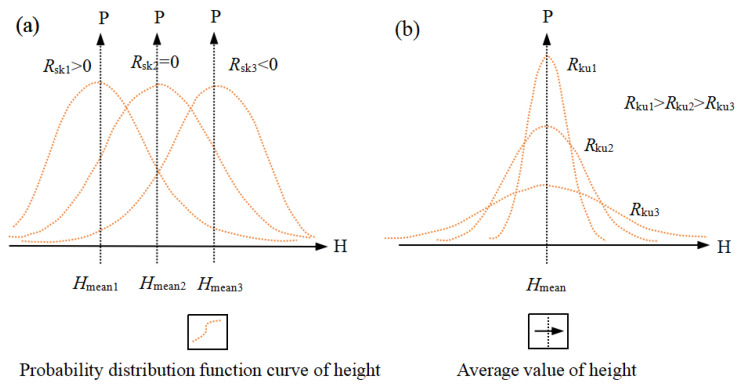
Shapes of (**a**) surface skewness (*R_sk_*) and (**b**) kurtosis coefficient (*R_ku_*).

**Figure 4 materials-16-05564-f004:**
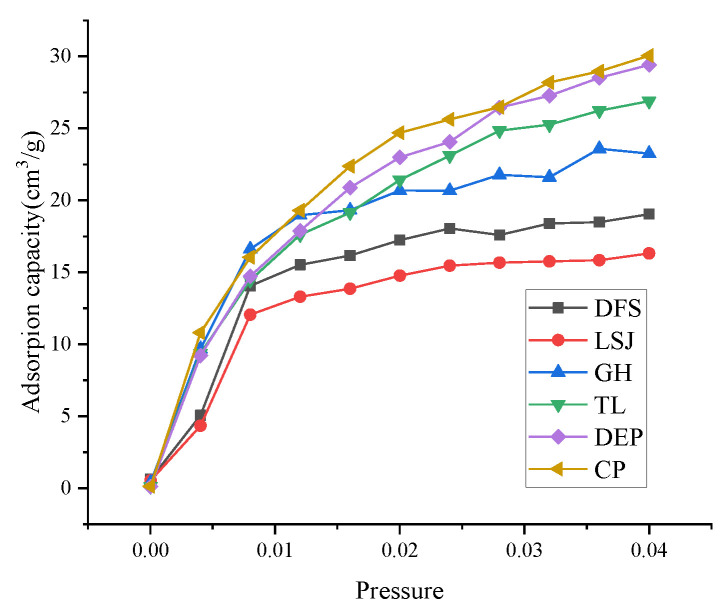
Adsorption–desorption curve obtained by LT-CO_2_-GA.

**Figure 5 materials-16-05564-f005:**
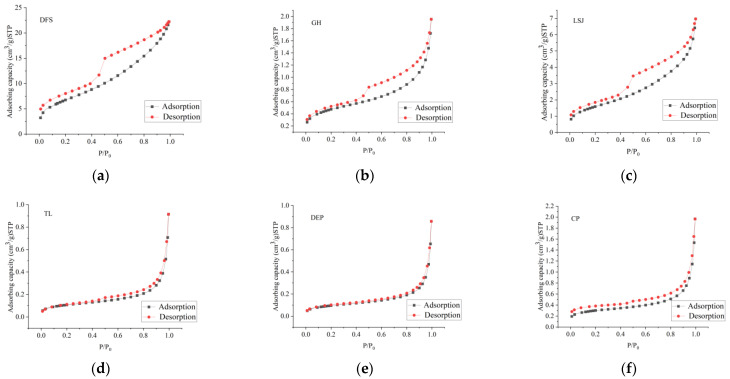
Adsorption–desorption curve obtained by LT-N_2_-GA. (**a**) DFS; (**b**) LSJ; (**c**) GH; (**d**) TL; (**e**) DEP; (**f**) CP.

**Figure 6 materials-16-05564-f006:**
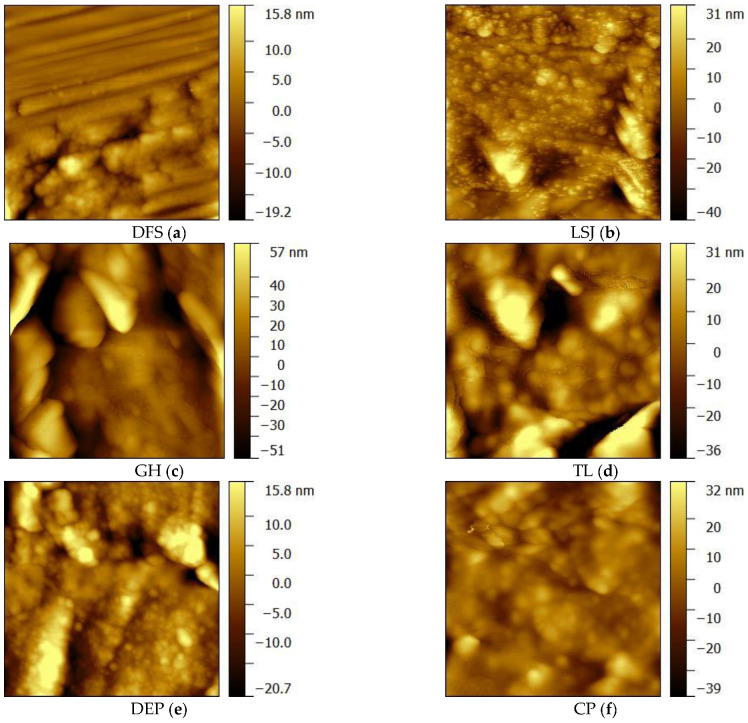
Coal microscopic morphology obtained by AFM.

**Figure 7 materials-16-05564-f007:**
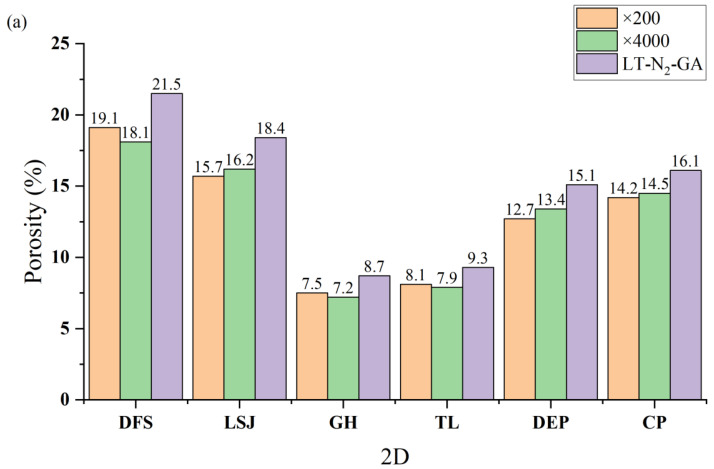
Comparison of the porosity histograms of coal samples obtained by different algorithms and magnifications: (**a**) the Threshold algorithm and (**b**) Chen’s algorithm.

**Figure 8 materials-16-05564-f008:**
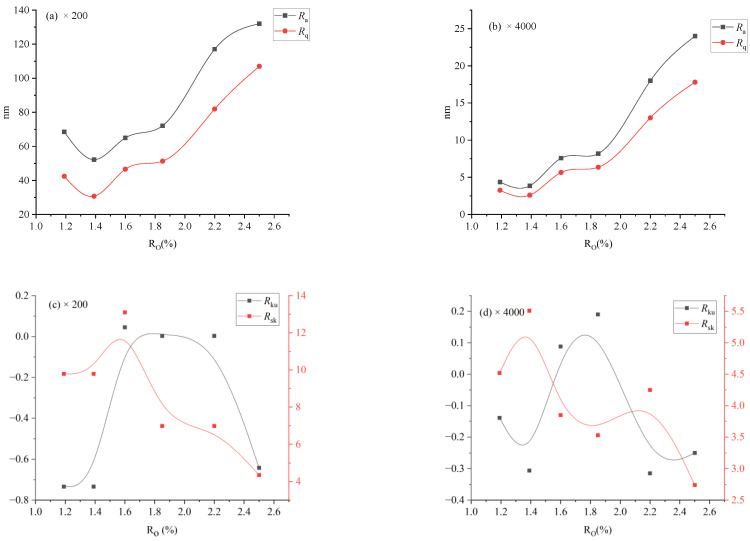
The relationship of surface roughness and *R*_o,max_.

**Figure 9 materials-16-05564-f009:**
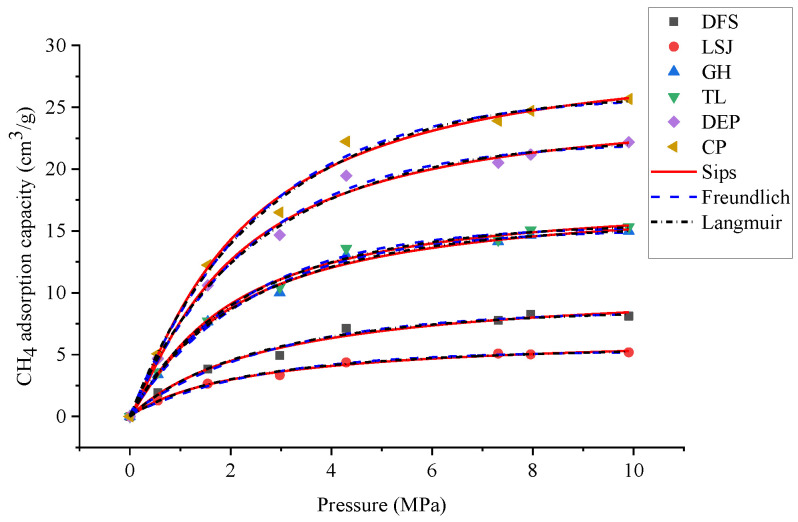
The adsorption isotherms simulated by Langmuir, Freundlich, and Sips models.

**Figure 10 materials-16-05564-f010:**
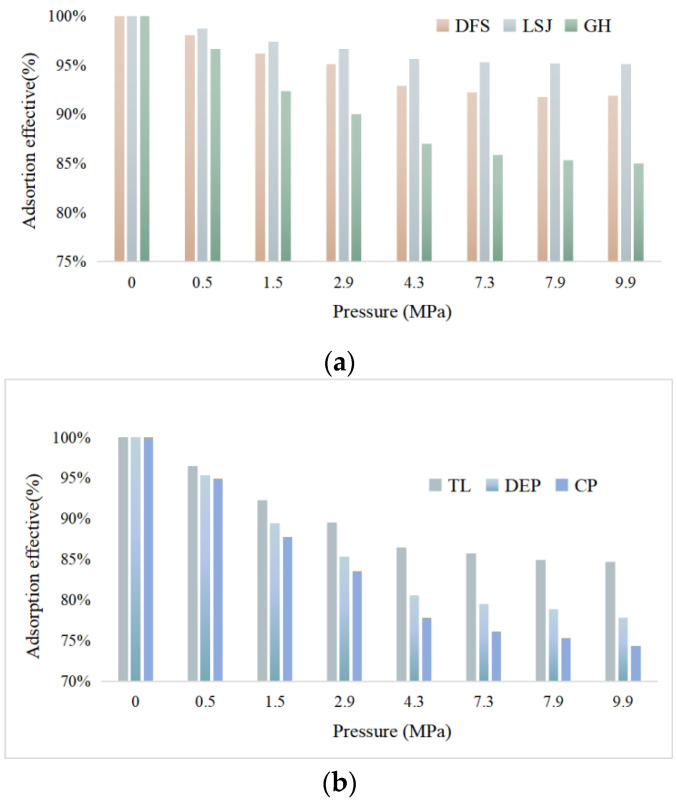
The adsorption effectiveness of samples. (**a**) Adsorption effective of DFS, LSJ and GH; (**b**) Adsorption effective of TL, DEPand CP.

**Figure 11 materials-16-05564-f011:**
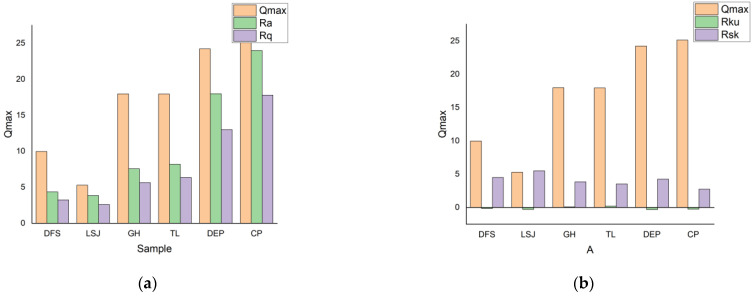
The relationship of surface roughness and *Q*_max_. (**a**) The relationship of *R*_a_, *R*_q_ and *Q*_max_; (**b**) The relationship of *R*_ku_, *R*_sk_ and *Q*_max_.

**Table 1 materials-16-05564-t001:** Proximate analysis and vitrinite reflectance results.

Sample ID	Proximate Analysis (%)			*R*_o,max_ (%)	Classify
	*M* _ad_	*A* _d_	*V* _daf_		
DFS	3.76	7.14	33.70	1.19	High volatile bituminous coal (HVB)
LSJ	4.78	10.98	32.30	1.29	High volatile bituminous coal
GH	0.62	7.88	23.77	1.37	Middle volatile bituminous coal (MVB)
TL	0.66	7.72	23.76	1.58	Middle volatile bituminous coal
DEP	1.08	6.49	16.00	1.89	Low volatile bituminous coal (LVB)
CP	0.75	10.12	13.82	1.97	Low volatile bituminous coal

*M*_ad_ = moisture content; *A*_d_ = ash content; *V*_daf_ = volatile content; *R*_o,max_ = maximum oil vitrinite reflectance.

**Table 2 materials-16-05564-t002:** Micropore structure parameters of six samples.

Sample ID	*SSA* (m^3^/g)	*PV*_min_ (cm^3^/g)
DFS	43.354	0.019
LSJ	32.027	0.012
GH	52.813	0.014
TL	58.049	0.019
DEP	64.891	0.023
CP	69.924	0.026

*PV*_min_ = pore volume of micropore.

**Table 3 materials-16-05564-t003:** Mesopore and macropore structure parameters of six samples.

Sample ID	SSA (m^3^/g)	*PV*_min_ (cm^3^/g)	*PV*_mes_ (10^−3^ ∗ cm^3^/g)	*PV*_mac_ (cm^3^/g)
DFS	0.675	0	2.359	0.030
LSJ	0.494	0	2.005	0.029
GH	0.702	0	2.436	0.029
TL	0.754	0	3.250	0.036
DEP	0.921	0	3.997	0.026
CP	1.056	0	4.231	0.020

*PV*_mes_ = pore volume of mesopore; *PV*_mac_ = pore volume of macropore.

**Table 4 materials-16-05564-t004:** The pore parameters tested by AFM calculated by the Threshold algorithm for coal samples (×4000/×200).

Sample ID	Proportion of Pore Volume (%)			Number of Pores	Maximum Pore Size (nm)	Minimum Pore Size (nm)
	Micropore (0–2 nm)	Mesopore (2–50 nm)	Macropore (>50 nm)			
DFS	26.89/23.09	51.25/52.62	21.86/24.29	1056/264	220.4/219.2	0.18/0.35
LSJ	23.24/26.51	30.63/29.36	46.13/44.13	792/196	287.3/288.6	0.20/0.39
GH	11.51/10.22	44.92/45.84	43.57/43.94	756/182	415.5/414.7	0.19/0.40
TL	14.23/13.43	46.83/45.78	38.94/40.79	687/188	368.1/370.1	0.18/0.37
DEP	32.47/35.64	43.18/42.39	24.35/21.27	521/132	400.5/401.3	0.19/0.39
CP	36.46/37.36	41.33/43.64	22.21/19.00	494/127	430.1/429.9	0.17/0.36

**Table 5 materials-16-05564-t005:** The pore parameters tested by AFM calculated by Chen’s algorithm for coal samples (×4000/×200).

Sample ID	Proportion of Pore Volume (%)			Number of Pores	Maximum Pore Size (nm)	Minimum Pore Size (nm)
	Micropore (0–2 nm)	Mesopore (2–50 nm)	Macropore (>50 nm)			
DFS	34.3/34.9	5.8/5.6	60/59.5	1220/288	234.5/235.3	0.12/0.28
LSJ	27.9/27.9	4.6/4.6	67.5/67.4	1019/254	294.2/293.7	0.14/0.30
GH	31.7/33.1	4.9/5.0	63.4/61.9	841/215	431.6/429.1	0.15/0.31
TL	34.7/35.1	6.3/6.5	59/58.5	726/192	375.3/378.4	0.14/0.30
DEP	45.5/45.5	7.1/7.1	47.4/47.4	687/167	418.7/416.6	0.13/0.29
CP	52.8/53.7	8.1/7.5	39.1/38.4	563/154	452.1/448.8	0.14/0.27

**Table 6 materials-16-05564-t006:** The surface roughness of samples.

Sample	Magnification	*R* _sk_	*R* _ku_	*R*_q_ (nm)	*R*_a_ (nm)
DFS	×200	−0.73	9.79	68.5	42.4
	×4000	−0.14	4.52	4.36	3.25
LSJ	×200	−0.69	9.61	52.2	30.7
	×4000	−0.31	5.51	3.85	2.60
GH	×200	0.04	13.1	65.0	46.6
	×4000	0.08	3.85	7.57	5.65
TL	×200	0.01	6.98	72.1	51.3
	×4000	0.19	3.53	8.19	6.36
DEP	×200	−0.57	6.70	117	81.9
	×4000	−0.31	4.25	18.0	13.0
CP	×200	−0.64	4.35	132	107
	×4000	−0.25	2.74	24.0	17.8

**Table 7 materials-16-05564-t007:** The adsorption parameters of samples.

Isotherm Models	Parameters	DFS	LSJ	GH	TL	DEP	CP
Langmuir	*Q* _max_	10.600	5.800	18.450	18.450	26.110	26.110
	*P* _L_	2.790	1.800	1.940	1.950	1.840	1.980
	*R* ^2^	0.979	0.978	0.977	0.980	0.976	0.979
Freundlich	*P* _L_	2.640	1.720	1.850	1.840	1.750	1.890
	*R* ^2^	0.981	0.984	0.986	0.984	0.988	0.989
Sips	*Q* _max_	9.980	5.300	17.980	17.950	24.230	25.120
	*P* _L_	2.620	1.690	1.820	1.820	1.730	1.860
	*R* ^2^	0.992	0.991	0.993	0.990	0.991	0.994

## Data Availability

Not applicable.

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
