# Peer review of "Three-Dimensional Pore Structure Characterization of Bituminous Coal and Its Relationship with Adsorption Capacity"

_materials, 2023, doi:10.3390/ma16165564_

Round 1
Reviewer 1 Report
Comments are attached

Quality of English is low.
Author Response
Reviewer1:
But, overall the manuscript is very poorly written especially in section 4 of the manuscript (4. Discussion) and Conclusion section, and therefore would need extensive re- writing before it could be considered for publication. The general impression is that the manuscript is written in a rush, with many grammatical mistakes.
There many others completely muddled phrases with a lot of grammar and language mistakes in the text and in the Figures.
Sections 4. 'Discussion' and 5. 'Conclusion' have been thoroughly revised.
I have focused on enhancing the grammar, sentence structure, coherence, and overall fluidity of the content to meet the stringent standards of academic publishing.
1.Page 6: Figure3 is not clear and it should be redone
Thank you for your feedback. I had worked on improving the image quality and readability, ensuring it effectively communicates the necessary information.
2.Page7: The authors claim that: “FIGURE shown the isothermal adsorption curves of CO2 on all samples” but the figure is missing!
I apologize for the oversight and appreciate your patience. Indeed, Figure 4, which is intended to show the isothermal adsorption curves of CO2 on all samples, seems to be absent from the current version of the manuscript.
3.Thereare many sentences and phrases with grammar and language mistakes in the text: Line 166; Line 232; Line 299;
I had promptly revised these specific lines, along with the rest of the manuscript, to correct any grammatical or language errors.
4.There are completely muddled phrases with a lot of grammar and language mistakes in the text: - lines 303-308; lines 311-313; lines 335-338; lines 350-355; lines387- 392.
I had promptly revised these specific lines, along with the rest of the manuscript, to correct any grammatical or language errors.
5. Line 365: The authors claim that: “The CH4 adsorption volume( VL) is fitted by Langmuir model shown in section 2.3.1 show in Table 7”, but section 2.3.1 is missing! By the way, from section 2.2.3. it is jumping to section 2.33. (D roughness)
My apologies for the confusion. The mention of section 2.3.1 appears to be a mistake. The correct reference should be section 2.4.2.
6. Conclusion chapter has muddled phrases with grammar and language mistakes, and it is very short compared to the text of the
'Conclusion' have been thoroughly revised. (1)Upon conducting a comparative analysis of AFM images across various algorithms and scales, we posit that the pore calculation results derived from the 3D algorithm at ×4000 magnification are more accurate than those obtained through other algorithms. These results exhibit greater resemblance to the LP-CO2/N2-GA findings.Chen's algorithm discerned a larger number of pores than the Yen algorithm. For example, in the case of DFS, the numbers were 1220 (×4000) versus 1056 (×4000). Furthermore, Chen's algorithm uncovered more micropores. The porosity determined by the 3D algorithm outperformed that of the Threshold algorithm and was closer to the LP-N2-GA results.When observed at a magnification of ×4000, more pores were identified than at ×200 (DFS: 1056 vs. 264 using the Threshold algorithm). However, the porosities observed at magnifications of ×200 and ×4000 nm were similar, rendering the effect of magnification inconsequential.
(2)The second coalification transition exerts a significant impact on the coal's pore structure. Over time, the structure evolves from linear parallel cracks and cylindrical pores to microcracks and wedge-shaped pores. Simultaneously, coal's pore volume and surface roughness initially decline before escalating, correlating with the coal rank.Ra and Rq decrease linearly with the increase, the Rku value increased, Rsk value is greater than 0 in the early stage and gradually less than 0. This variation is predominantly attributed to the transformation of the coal's macromolecular structure.
(3)Surface roughness significantly impacts the gas adsorption capacity of samples. A more pronounced fluctuation in coal structure corresponds to a higher gas storage capacity. As a result, Rsk and Rku serve as reliable indicators of gas adsorption potential. Furthermore, smaller Ra and Rq values, indicative of a smoother coal surface, result in diminished friction between the gas and coal surface, thereby enhancing gas adsorption.Author Contributions: All authors contributed equally to this work.

Reviewer 2 Report
The article requires major revisions, specifically regarding the following topics and paragraphs:
It is requested to improve the abstract and introduction by indicating the purpose of the work and referring to the state of the art by comparison with other bibliographic sources;
In materials and methods it is required to report the reference standard methods and the protocols used for the proximate analysis.
The ash values ​​for the LSJ and CP samples are high, how do you explain this result?
It is requested to check the abbreviations within the tables, the acronym is not always expressed by indicating the term in full.
There is no statistical evaluation of the data obtained, it is requested to integrate the discussions with this elaboration.
Moreover, a paragraph should be inserted in which the effective capacity of adsorpion is demonstrated.
It is necessary to specify in the text what is the purposes of the study, explaining why the evaluation carried out is useful for the purposes of applying the investigated matrix
Moderate revision
Reviewer 3 Report
This manuscript reports Three dimensions pore structure characterization of bituminous and its relationship with adsorption capacity. Following minor revisions should be made:
1. Avoid numbering in Abstract. There are some continuous references in the experimental section. This indicates the lack of methodology.
2. Moderate editing of English language required
3. It will be better to explain the porosity of LT-N2-GA by BJH.
4. Adsorption isotherms and kinetics should be studied for several models. Refer:
Chemical Engineering Journal, Volume 417, 1 August 2021, 129312
1. Moderate editing of English language required
Round 2
Reviewer 1 Report
Comments are attached.

Quality of English language is good.
Author Response
- Page 8: The authors have added a new Figure which should be Figure 4, but the caption is wrong (line 244). The caption for this figure is identical with the one of the next figure (line 284).
The “Figure 5” in line 244 is modified to “Figure 4”.
- 2. Thechapter3 numberappears twice in the text: 4.3. Surface roughness evolution on second coalification jump (Line 419) and 4.3. Effect of 3D pore structure on CH4 adsorption capacity (Line 440)
The “4.3. Effect of 3D pore structure on CH4 adsorption capacity” (Line 440)
is modified to “4.4. Effect of 3D pore structure on CH4 adsorption capacity” .

Reviewer 2 Report
The integrations in the text are correct.
It is advisable to add a paragraph on the statistical elaboration of the results.
Review spaces, captions, and article text formatting.
Moderate revision
Author Response
1 It is advisable to add a paragraph on the statistical elaboration of the results.
AFM, employing Chen's algorithm and a magnification of ×4000, can accurately analyze the 3D pore structure of bituminous coal. Based on this, the range of pore quantity in bituminous coal is found to be 563-1220, with the maximum value of CP and minimum of DFS; The range of maximum pore size is 234.5-234.5 nm, while the range of minimum pore size is 0.12-0.15 nm, these values show minimal variation with respect to coal rank; The variation range of porosity is 7.4% to 21.1%, with GH having the minimum value; Rsk ranges from -0.31 to 0.19, and Rku ranges from 2.74 to 5.51, with weak regularity in their variations. The range of Rq is 3.85-3.85, and Ra ranges from 2.60 to 17.8, with LSJ having the minimum value and CP having the maximum value. Among the different adsorption models, the Sips model exhibits the best fitting performance. The Qmax values for CH4 adsorption range from 5.30 to 25.12 cm3/g. The ordering of adsorption capacity from highest to lowest is CP > DEP > TL > GH > DFS > LSJ, which aligns with the observed ordering of Qmax.
2 Review spaces, captions, and article text formatting.
The spaces, captions, and article text formatting are modified.
For example: (1)The “Figure 5” in line 244 is modified to “Figure 4”.
(2)The “4.3. Effect of 3D pore structure on CH4 adsorption capacity” (Line 440) is modified to “4.4. Effect of 3D pore structure on CH4 adsorption capacity” .
